# Effects of the Mediterranean Diet on Morbidity from Inflammatory and Recurrent Diseases with Special Reference to Childhood Asthma

**DOI:** 10.3390/nu14050936

**Published:** 2022-02-22

**Authors:** Fernando M. Calatayud-Sáez, Blanca Calatayud, Ana Calatayud

**Affiliations:** Child and Adolescent Clinic “La Palma”, C/Palma 17, bajo A, 13001 Ciudad Real, Spain; blanca.calatayud@gmail.com (B.C.); anacalatayud94@gmail.com (A.C.)

**Keywords:** inflammatory disease, recurrent disease, recurrent colds, recurrent acute otitis media, recurrent acute rhinosinusitis, otitis media with effusion, persistent nasal obstruction, childhood asthma, Mediterranean diet, dietary intervention, nutritional therapy

## Abstract

**Objective:** For 15 years, we have been working with a nutritional programme based on the traditional Mediterranean diet (TMD) to complete the treatment of inflammatory and recurrent diseases (IRD), such as childhood asthma. The objective of this study is to verify the effects of TMD in the prevention and treatment of IRD by measuring the incidence of infant morbidity over 8 years. **Material and Methods:** The number of patients who suffered from IRD each year (just before the pandemic) was determined, as well as the frequentation and the percentage of scheduled and on-demand consultations. **Results:** The incidence of infant morbidity decreased as they were incorporated into a TMD, and we observed a progressive disappearance of IRD. At the beginning of the study, 20% of the patients had been diagnosed with some type of IRD. At the study’s end, the prevalence of IRD decreased to less than 2%, and the use of drugs and surgical interventions decreased markedly. **Conclusions:** A diet based on the TMD reduces the incidence of infant morbidity and contributes to the disappearance of IRD, whereas some non-traditional foods with high antigenic power could be involved in the appearance of IRD.

## 1. Introduction

The most frequent reason for consultation in primary care paediatrics is inflammatory diseases, which are repeated during the winter months once the school year has begun. These are termed inflammatory recurrent diseases (IRD), which, as their name indicates, have an inflammatory etiological basis with recurrence being a defining characteristic. The most common ones affect the respiratory area, such as recurrent colds, recurrent acute otitis media (RAOM), recurrent acute rhinosinusitis (RARS), otitis media with effusion (OME), persistent nasal obstruction (PNO), and childhood asthma. Conservative and symptomatic treatment is usually sufficient to control the acute phase of these diseases. However, sometimes antibiotics and anti-inflammatories must be used, but these drugs are not effective in preventing new episodes, which induces the use of these drugs repeatedly. 

In terms of inflammatory triggers, the effects of a Western lifestyle have been trivialized, and circumstances that seem detrimental to the maintenance of children’s health have not been considered, such as environmental pollution, the generalisation of chemical products, and the abandonment of the traditional diet. Increasingly, studies have demonstrated the anti-inflammatory effects of the Mediterranean diet [1,2,3] and its ability to control immune-based diseases, which has allowed us to develop the hypothesis that recurrent inflammatory episodes of the mucous membranes are closely related to the abandonment of the traditional diet. For about 15 years, we have been working with a nutritional programme based on the traditional Mediterranean diet (TMD), and we have applied it to all patients who have presented with some type of IRD. We have conducted studies on the effects of TMD on recurrent colds and their common bacterial complications (RCBC) [4], such as RAOM [5], RARS [6], OME [7], PNO [8], and childhood asthma [9]. In all of them, we have concluded that the TMD has potent anti-inflammatory effects and is effective in the control and treatment of IRD. Given the achievement of these satisfactory results, we extended the TMD to all healthy children from birth and verified that the application of the TMD to the entire child population leads to lower incidence of acute diseases and the near disappearance of IRDs [10]. We have, therefore, observed that TMD has an important therapeutic action in the control of IRD, as well as a notable preventive action against common acute childhood diseases. The objective of this study was to assess the effects of the progressive incorporation of the TMD on the frequency of use and the incidence of morbidity due to IRD, with special reference to childhood asthma, in all patients who attended paediatric consultation of primary care over 8 years. 

## 2. Material and Methods

All patients in our paediatric primary care pool (approximately 1200 patients) were asked to participate in the study from the beginning of 2011 to the end of 2018. Initially, all those who had some type of IRD were included, and later all children were included from birth. All were evaluated by the paediatrician and the nutritionist at 4 and 12 months after the initial visit, after which an annual review was conducted.

All patients who presented some type of IRD were included and progressively incorporated into the TMD through a nutritional education programme, and their evolution was assessed through prospective before–after (pre/post-test) comparison studies lasting 1 year for each patient. Subsequently, we progressively expanded the nutritional programme to all healthy children from birth, for which a single-group intervention was carried out in which we evaluated the differences in the results according to the degree of adherence to the TMD recommendations [10]. In recent years, practically all of our child population and their families were aware of the TMD and apply it to a greater or lesser extent in their daily diet. The children who were captured shortly after birth were immediately incorporated into the nutritional programme, promoting breastfeeding from the beginning. An explanatory diagram is shown in Figure 1.

Concerning attendance, patients were divided into two groups: (a) those who attended due to acute illness and in haste, and (b) those who attended the scheduled control and without haste. Bureaucratic consultations were not considered. Concerning morbidity, we have defined IRD as the repetition of the same inflammatory pathology in more than three episodes in 6 months, or four episodes in 1 year. Regarding IRDs, the following variables were considered: RCBC, RAOM, RARS, OME, PNO, and RW. Data were also collected on the different drugs prescribed and emergency care, assessed by person and year. In recent years, those children who had developed some type of IRD underwent blood tests with inflammatory markers and the Histamine Reaction to Food Antigens (HANA) test was performed [11]. 

The criteria to define each of the IRDs were based on the “expert groups” of each of the nosological entities. An episode of upper respiratory tract infection (URTI) was defined when patients had fever over 38 °C, sore throat, runny nose, shortness of breath through the nose, and a cough [12]. Acute otitis media (AOM) was defined as a process of tympanic bulging, caused acutely during an URTI and with inflammatory signs or symptoms such as otalgia, eardrum redness, or otorrhea [13]. Acute rhinosinusitis (ARS) was defined as a prolongation of URTI symptoms for more than 10 days, with nasal obstruction, thick mucus secretion, and persistent cough predominantly at night [14]. OME was defined as bilateral tubal obstruction lasting for more than 3 months, or unilateral for more than 6 months [15,16]. PNO was defined as persistent difficulty breathing adequately through the nose and associated symptoms such as snoring, mouth breathing, sleep apnea, drowsiness, and difficulty swallowing [17]. RW was defined when three or more episodes of wheezing and coughing occurred, in a context in which the diagnosis of asthma is the most likely, after ruling out less frequent processes [18]. Allergy tests were only performed on suspected allergy patients. The clinical evolution of the patients was evaluated using a specific questionnaire for each IRD, designed to be answered by the parents or guardians (Appendix A). Weight, height, and body mass index were measured to evaluate weight–stature evolution [19].

Parameters for adherence to the TMD: We have used the nutritional program “Learning to eat from the Mediterranean”, which we have explained in detail in previous articles [10]. Our aim was that all patients would have a broad knowledge of the characteristics of the TMD and an ability to follow it. To assess adherence to the TMD, we used the KidMed test [20,21], the TMD test [19], and the TMD-Infant test [10]. The KidMed test is a classic tool for measuring adherence to the Mediterranean diet (Appendix B), although the TMD Test (Appendix C) seems to be similar, as does the TMD-Infant test for evaluating newborns and infants (Appendix D). The traditional Mediterranean diet is characterized by the presence of fresh and seasonal foods, which must be consumed in the short-term since they easily spoil. In particular, it is worth noting the abundant consumption of fruits, vegetables, legumes, and whole grains. Olive oil, seeds, and nuts are the main sources of fat. There is a low consumption of proteins and fats of animal origin. Dairy products are usually fermented and come from goats and sheep, and fish is consumed in moderation. It is important to note that pre-cooked and industrial foods, which have undergone profound organoleptic modifications, are not typical of the TMD [22]. In Appendix E, we present the decalogue that the Mediterranean Diet Foundation proposes to us through its website [23]. The TMD has been proclaimed by UNESCO as ‘cultural heritage and intangible asset of humanity’ [24]. In Table 1, we delineate the main differences between the Mediterranean diet and the diet that has been imposed by “Western civilization”.

Laboratory evaluation: Only children who had some type of IRD were evaluated using general biochemistry, together with immunoglobulins, total and specific IgE to respiratory and food allergens, inflammatory markers (high sensitivity C-Reactive protein (hs-CRP), tumor necrosis factor Alpha (TNF-alpha), interleukin-1 (IL-1), interleukin-6 (IL-6), complement factor-3 (C-3), complement factor-4 (C-4)), and test of histamine response to food antigens or the HANA test [11]. For the patients who dropped out of the study or did not want to participate in the nutritional programme, the reasons given were noted in their records (transfer, social difficulties in following the diet, and conflict with the limitation of some foods). Sample size and statistical analysis: A non-probabilistic sample was taken of all patients (guardians/families) who consecutively consented to participate in the study. Descriptive analysis was performed with measures of central tendency and dispersion for quantitative variables (mean and standard deviation), and frequency distribution for qualitative ones. For the comparison of quantitative variables between two groups, the Student’s t-test (or Mann–Whitney U-test in the case of a non-normal distribution) was used, establishing the level of statistical significance at *p* < 0.05. For the analysis of the results, the statistical package SPSS 15.0 was used.

## 3. Results

Most of the families followed our nutritional recommendations, although a small percentage, less than 8%, reported problems with the diet. The greatest difficulties had to do with the lack of time to make changes to the menu and introducing homemade meals, and occasionally due to a lack of agreement to limit some foods. The results obtained were similar in both sexes, so they are reported together. The pondero-statural development was adequate and proceeded as planned. However, it should be noted that overweight and obesity decreased throughout the study. Table 2 shows the number of patients diagnosed with IRD each year and also the number of overweight and obese patients after the progressive introduction of the TMD in the paediatric population.

Acute illness consultations with haste decreased as the patients entered the TMD programme while maintaining the number of scheduled control and without haste consultations, so the percentage of attendance at both consultations was equal. Table 3 shows the percentage of acute illness in haste and scheduled control without haste visits between 2011 and 2018.

Table 4 shows the average number of acute IRD episodes in the year prior to diagnosis and the following year after starting the application of the nutritional programme; IRD episodes per child and year were assessed. The times that patients went to emergency services are also shown, as well as the antibiotic treatment cycles they received during their inflammatory processes, and symptomatic treatments such as paracetamol, saline sprays, anti-inflammatories, or expectorant mucolytics.

Following the introduction of the nutritional programme, we observed greater adherence to the TMD as evidenced by the results of the KidMed test and the TMD test, before and after applying the intervention (Figure 2). 

At the beginning of the program, 70% of the patients obtained an optimal score on the KidMed test (above 8 points). This rose to 90% by the end of the study. In the TMD-test, only 10% of patients obtained an acceptable score at the beginning of the study. This progressed satisfactorily, until reaching an average score classified as optimal (above 16 points) (Figure 3). We carried out both tests, because in previous studies we observed that the KidMed test was not sensitive to the changes that we promoted in infant feeding, while the TMD test did reflect these proposed changes.

In healthy children who accessed the TMD from birth, a low incidence of acute inflammatory diseases and IRD was observed, and this was more notable in those who were breastfed (group 1) compared to those who drank adapted milk (group 2). Compliance with the rest of the TMD parameters was similar in both groups. Both groups evolved satisfactorily with a notable decrease in IRD, in-demand consultations due to illness, and the need to go to the emergency room or require medication treatments. The number of RW attacks or childhood asthma was lower and clinically relevant in group 1, with an incidence of almost half the rate of group 2. The number of drugs used by patients with RW in group 1 was less than half of the number used by patients in group 2 (Table 5). 

In the group of patients to whom we applied the TMD from birth, a total of 38 IRD episodes appeared in the first two years of life, with 61% episodes of RW or childhood asthma (10 in Group 1 and 28 in Group 2). All patients with IRD developed elevated inflammatory markers, especially TNF-alpha. The HANA test marked foods suspected of promoting a non-IgE-mediated inflammatory response. Cow’s milk was the food antigen most frequently indicated by the HANA test, followed in frequency by beef and pork. Among vegetables, a sensitivity to wheat stood out. After restricting the foods indicated by the HANA test and adapting the diet, we re-assessed the clinical and analytical evolution at 6 months. Overall, the response was satisfactory, with RW or childhood asthma and other IRDs disappearing in those patients who followed our indications (70%) 3–4 weeks after starting dietary restructuring. The rest of the patients (30%) reported difficulties following the exclusion diet, and we proposed they receive the conventional treatment.

## 4. Discussion

The incidence of infant morbidity progressively decreased throughout the 8 years of the study. Supporting the hypothesis, we observed that a quality diet based on the TMD decreased infant morbidity and, as a consequence, a progressive disappearance of IRD. The application of the TMD has not only been effective in treating IRD, as we have found in previous studies, but its application from birth is effective in preventing its appearance. This was more evident in the group of patients who had higher affinity for the TMD. At the beginning of the study, more than 20% of our patients had been diagnosed with some type of IRD, and its prevalence decreased to 2% in the last year. Consistent with the above results, the use of drugs and surgical interventions decreased markedly. The degree of satisfaction of the families with the programme was high. These circumstances have meant that on-demand consultations for illness have been progressively fewer in number than scheduled consultations, something uncommon in primary care paediatric consultations. At present, almost 50% of paediatric consultations are scheduled, and on-demand consultations continue to decline.

From a total of 244 IRD episodes diagnosed in 2011, only 24 were diagnosed in 2018. Each year, fewer patients were diagnosed with little change in the total number of patients in the paediatric space and without other factors, such as age, starting time, schooling, or the number of siblings having an impact on the results. This can only be explained by an increase in the effectiveness of the patients’ defensive system, either by an early maturation of the immune system or stabilisation and balance of the inflammatory system. 

The immune system tends to get stronger with age and IRDs tend to disappear spontaneously. However, with nutritional intervention, we have obtained beneficial results, which seem to indicate a strengthening of the immune system, since IRD episodes decreased rapidly and there was less need for pharmacological treatments or surgery.

RCBC are one of the most frequently seen conditions in primary care. During our study, RCBC progressively decreased from 46 patients to less than 5 per year. In addition, we observed a lesser effect on general symptoms and quicker recovery [4]. AOM is one of the most common reasons for consultation (25–40%) and is the leading cause of antibiotic prescription in childhood. Its incidence dropped considerably after the application of the TMD. RAOM, which can affect a third of the child population and is one of the most frequent recurrent diseases in paediatrics, gradually reduced in our study from 27 patients a year to less than 3 per year [5]. ARS is another cause for the increased prescription of antibiotics in childhood and can affect 10% of children, many of whom end up suffering from RARS. In our study, we went from having 40 patients with RARS to less than 3 per year [6]. These data have been consistent with mass vaccination for haemophilus influenza and streptococcus pneumonia, which could have influenced the achievement of these results. However, some publications do not show that this fact has reduced the incidence of these diseases in the same proportion as in our study [25]. Ringing of the ears, which is usually associated with recurrent catarrhal processes and nasal obstruction, was resolved more easily, and tympanometry returned to normal within a few weeks for the patients in our study. Hence, OME has decreased from 24 patients to less than 2 per year [7]. Coinciding with recurrence and the permanent inflammatory state of the upper respiratory tract, many patients develop hypertrophy of the tonsils and adenoids, contributing to difficult nasal breathing. In our study, PNO decreased in incidence from 27 patients to less than 3 per year. We must emphasise that as a consequence of the improvement experienced in all IRDs, surgical intervention was not necessary, and most of the children preserved their tonsils, adenoids, and the integrity of the tympanic membrane. 

By reducing the consumption of processed foods and those with a high glycaemic index, together with an excess of foods of animal origin, we have also observed a decrease in the incidence of overweight and obesity [19]. This indicates that TMD, in addition to protecting the defensive system, can promote the establishment of an adequate metabolism [26]. Infant asthma (RW) in the subgroup that left breastfeeding early had an incidence similar to the national database (15%) [27]. In this group, the rest of the TMD completion parameters were similar, which makes us suspect that adapted cow’s milk could be closely related to the development of bronchial hyper-reactivity and, therefore, breastfeeding could offer important protection [28,29]. Patients who were fed breast milk had a lower incidence of RW or childhood asthma, despite which they maintained a somewhat high figure, compared to the evolution of other IRDs. It would have been interesting to have applied a TMD to mothers during pregnancy and lactation since antigenic substances from foods that are not typical of the TMD could have reached them through the placenta or breast milk [30]. Childhood asthma (RW) is generally triggered by viral infections which stimulate a previously pro-inflamed or hyper-reactive mucosa [31]. For this reason, inhaled corticosteroids are used for prolonged periods to dampen or control this undue inflammatory response [32]. Likewise, the food industry has introduced hypoallergenic milk on the market with a high degree of hydrolysis of cow’s milk proteins, based on evidence from studies that try to prevent childhood asthma [33]. 

At present, non-traditional foods have been massively incorporated into children’s diets, many of them of industrial origins such as ultra-processed and meat derivatives. Such foods can result in an antigenic overload of the immune system [34], which takes several years to resolve and translates clinically into defensive insufficiency in the lability of the inflammatory system and the hypertrophy of tonsils, adenoids, and lymph nodes. Most of the patients with IRDs studied in recent years had positive inflammatory markers, especially TNF-alpha, and to a lesser extent, hs-CRP, IL-1, IL-6, C-3, and C-4 [35]. Cow’s milk was the most frequent food antigen indicated by the HANA test, followed in frequency by beef and pork, and a sensitivity to wheat. Of the patients who tested positive for HANA, 70% were given a diet excluding foods to which they had sensitivity, resulting in the disappearance of IRDs. However, we believe it is necessary to expand these studies because we did not have a sufficient sample size to confirm these results. 

The effects of the Mediterranean diet on inflammation have recently been studied by numerous authors [1]. Adherence to the TMD reduces oxidative stress and chronic inflammation [2]. Individual nutrients have been shown to influence the development of inflammatory markers [36]. With the TMD, the pro-inflammatory actions of PAF (platelet activating factor), one of the most potent endogenous mediators of inflammation, can be favorably modulated [37]. Many of the products that are consumed in “Western civilization”, which have been processed to avoid fermentation, will favor the formation of unusual microflora that is alien to the specific human microbiota [38]. The excess of “antigenic load” caused, above all, by alteration of the microbiota and the increase in proteins of animal origin, could be the cause of the imbalance of the immune system and its excessive enlargement, ultimately causing pathological hyperplasia [39].

Our studies have been based on strengthening the body’s defensive system by recovering the traditional diet and eliminating foods that are not well-metabolized or are antigenic in nature. 

It has been proven that the intestinal microbiota induced by complementary feeding can cause changes that alter immune and metabolic mechanisms [40,41]. The components of these foods act on various immune cells and their effects are mediated by the intestinal immune system. An adequate immune response provides defenses to the host against infection, and the inhibition of immune responses can help control allergy and inflammation [42,43,44,45,46,47,48,49]. We believe that the introduction of non-traditional and antigenic baby foods may favor the alteration of inflammatory mechanisms, facilitating the appearance of IRD.

This study should be considered in light of its strengths and limitations. To assess adherence to TMD, we have preferred the TMD and TMD-B tests over the more classic KidMed test. This does not include some variables that we believe are important for the control of IRD. No difference was examined between refined cereals and whole grains, nor was the consumption of sugar or commercial fruit juices referenced. In general, the glycaemic index was not considered. In the lipid section, the consumption of saturated fats was not limited nor was its consumption evaluated, which does not allow us to detect an excess in the consumption of animal proteins. No assessment was made of the consumption of raw food, nor was the minimum amount to be taken specified. Serving sizes and schedules were also not considered. 

Our research study can be easily replicated in a pediatric consultation, provided that nutritionists are present. It is precisely the lack of nutritionists and the paucity of attention paid to nutrition that explains why the importance of diet in the development of acute diseases and IRD is ignored or obviated. We would have liked to carry out our studies with a control group. However, this was impossible because our child population was being taught to adhere to the TMD, and it did not seem appropriate to promote the standard diet of Western civilization, since we consider that it is precisely this diet that is the precipitating factor in the development of IRD. We also did not have sufficient funding to perform tests to measure the response of the inflammatory system and the modification of intestinal flora. It would also have been interesting to monitor expectant mothers and those who breastfeed their children.

In conclusion, the findings of this study suggest that many of the foods offered in “Western civilization” have a potentially high capacity to precipitate an undue inflammatory response by raising inflammatory markers, by precipitating the appearance of clinical symptoms compatible with alterations of the immune system. The limitation of suspicious foods and the application of the TMD have led to a considerable improvement in clinical symptoms and the normalization of inflammatory markers. We consider excess protein of animal origin and pre-cooked foods as the main suspect foods. Cow’s milk and other foods of animal origin could be implicated in the development of childhood asthma. The application of the traditional Mediterranean diet from birth, and the limitation of foods foreign to this diet and promoted by “Western civilization”, could significantly reduce the incidence of infant morbidity and the reduction of IRD, including childhood asthma.

## Figures and Tables

**Figure 1 nutrients-14-00936-f001:**
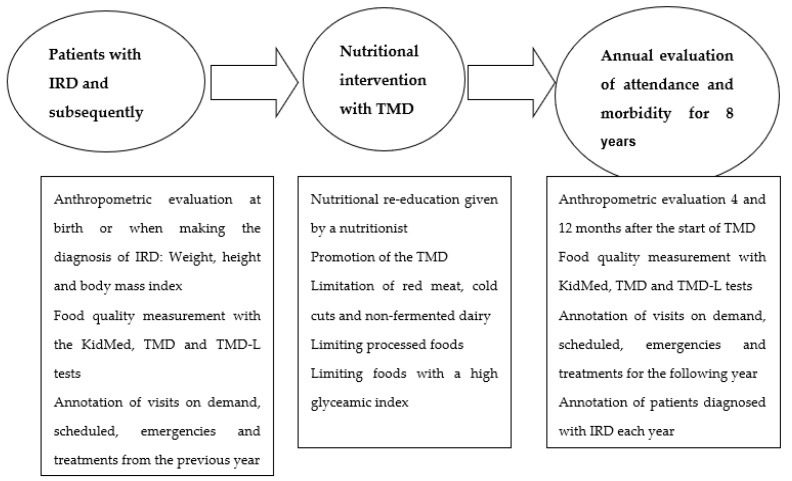
Study design diagram.

**Figure 2 nutrients-14-00936-f002:**
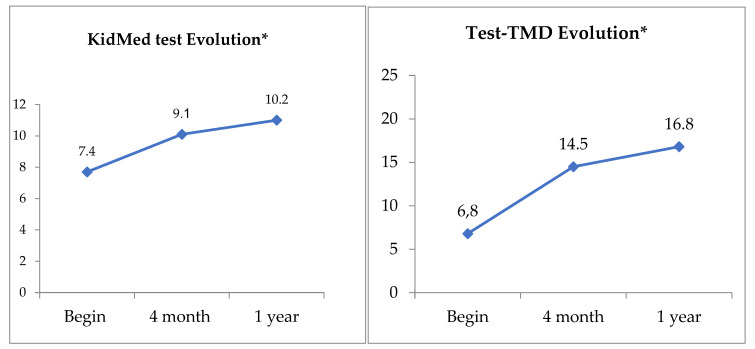
Evolution of the KidMed Test and Test-TMD. * KidMed and TMD index score.

**Figure 3 nutrients-14-00936-f003:**
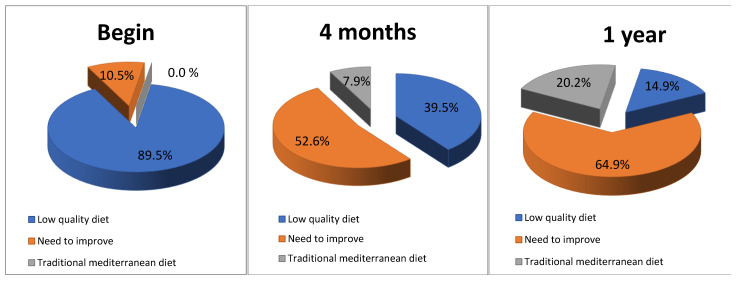
Evolution of diet quality, measured with the TMD Test.

**Table 1 nutrients-14-00936-t001:** Differences between the traditional Mediterranean diet and the “Western civilisation” diet.

Traditional Mediterranean Diet	Western Civilisation Diet
Breastfeeding	Adapted milk
Varied, seasonal fruit	Baby food jars and canned fruits
Vegetables and leafy vegetables	Baby food jars and canned vegetables and leafy vegetables
Pulses and non-processed nuts	Canned pulses and dried, fried, or salted nuts
Minimally processed and fermented whole grains	Refined, processed cereals with industrial fermenting agents
Fermented milk, principally goat’s and sheep’s	Whole, processed milks, mainly from cows
Occasional lean meat, in small quantities	High consumption of red, processed meats
Minimally processed, perishable, fresh and local foods	Nonperishable processed and ultra-processed foods
Homemade food	Pre-cooked food
Limits on products with added chemicals	Presence of chemical agents and enzyme disrupters

**Table 2 nutrients-14-00936-t002:** The evolution of the number of patients diagnosed with IRD each year after progressive introduction of the TMD.

Number of Children/Year	2011	2012	2013	2014	2015	2016	2017	2018
Recurrent colds with bacterial complications	46	36	19	18	16	6	5	4
Recurrent acute otitis media	27	12	10	9	9	4	2	2
Recurrent acute rhinosinusitis	40	24	12	11	10	5	3	2
Otitis media with effusion	24	16	10	9	6	4	3	1
Persistent nasal obstruction	27	12	11	11	5	2	1	2
Recurrent wheezing or childhood asthma	41	25	20	18	17	12	9	9
Overweight and obesity	19	13	12	11	9	4	3	3
Total	244	138	94	87	72	37	31	24

**Table 3 nutrients-14-00936-t003:** Percentage of visits for acute illness in haste and scheduled control without haste in each year.

Year	2011	2012	2013	2014	2015	2016	2017	2018
% Acute illness inquiries with haste	71%	65%	61%	59%	56%	52%	51%	51%
% Scheduled control and without haste consultations	29%	35%	39%	41%	44%	48%	49%	49%
Total number of consultations/days	31	30	28	25	20	18	16	16

**Table 4 nutrients-14-00936-t004:** Mean acute episodes of IRD during the year prior to diagnosis and after the application of the TMD.

	Year Before *	TMD Year *	*p **
Number of recurrent colds with bacterial complications	4.64 ± 0.70	0.70 ± 0.90	0.01
Number of episodes of recurrent acute otitis media	3.84 ± 0.73	0.48 ± 0.65	0.01
Number of episodes of recurrent acute rhinosinusitis	3.37 ± 1.21	0.32 ± 0.47	0.01
Degree of involvement in patients with persistent nasal obstruction: 0 (mild), 1 (moderate), 2 (severe)	1.92 ± 0.27	0.26 ± 0.05	0.01
% of patients with otitis media with effusion	100%	15%	0.01
Number of recurrent wheezing attacks or childhood asthma	4.73 ± 1.23	1.13 ± 0.71	0.01
Degree of involvement of IRD: 0 (mild), 1 (moderate), 2 (intense)	Between 1.3–1.8	Between 0.1–1.2	0.03
Emergencies per child and year	2.04 ± 0.79	0.25 ± 0.30	0.01
Antibiotic treatment cycles per child and year	3.51 ± 0.69	0.51 ± 0.33	0.01
Number of symptomatic treatments per child and year	6.87 ± 1.60	2.87 ± 0.93	0.01

* Mean and standard deviation. * Student’s *t*-test for independent groups was used to derive *p*-values.

**Table 5 nutrients-14-00936-t005:** Mean number of acute episodes and treatment in children less than two years of age for infectious diseases and childhood asthma in the two cohorts of healthy children.

Incidence per Patient in Two Years	Group 1	Group 2	*p **
Recurrent colds with bacterial complications	0.69 ± 0.96	1.22 ± 1.13	0.015
Recurrent acute otitis media	0.37 ± 0.68	0.51 ± 1.04	0.41
Recurrent acute rhinosinusitis	0.13 ± 0.34	0.20 ± 0.56	0.52
Recurrent wheezing or childhood asthma	1.15 ± 2.11	1.94 ± 2.14	0.06
Antibiotic treatment	0.44 ± 0.80	0.61 ± 0.77	0.29
Inhaled corticosteroids	0.75 ± 1.31	1.65 ± 1.95	0.17
Oral corticosteroids	0.08 ± 0.33	0.25 ± 0.65	0.08
Inhaled bronchodilators	1.06 ± 1.93	1.92 ± 2.17	0.033

* Group 1: breastfeeding + DMT. Group 2: adapted milk + DMT. The data are presented as means with their standard deviations. * Mean and standard deviation. Continuous data are shown as means ± S.D. and categorical as percentages. Student’s *t*-test for independent groups was used to derive *p*-values.

## Data Availability

Not applicable.

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
