# Peer review of "Effects of the Mediterranean Diet on Morbidity from Inflammatory and Recurrent Diseases with Special Reference to Childhood Asthma"

_nutrients, 2022, doi:10.3390/nu14050936_

Round 1

Reviewer 1 Report

To the Authors

Brief Summary

The objective of this study was to investigate the efficacy of the Traditional Mediterranean Diet (MD) in the prevention and treatment of inflammatory and recurrent diseases (IRD) including pediatric asthma, on the incidence of infant morbidity over a period of 8 years. Data analysis showed that adherence to the MD resulted in a reduction of infant morbidity, IRD prevalence, medication use, and surgical intervention. Furthermore, antigenic-producing foods such as cow’s milk, wheat, pork, and beef, typical of a Western diet were related to IRD and asthma. The authors conclude that a diet based on the MD reduces the incidence of infant morbidity and contributes to the disappearance of IRD, whereas some non-traditional foods with high antigenic power could be involved in the appearance of IRD.

Indeed, the high prevalence of inflammatory diseases including asthma in pediatrics, and the burden of associated morbidities is a major concern in public health.  Given that adherence to conventional pharmacological treatment is suboptimal, especially in children suffering from asthma, it is worth investigating non-pharmacological approaches that could reduce asthma symptomology and subsequently burden and economic costs of this disease.  To date, systematic reviews and meta-analyses of observational studies suggest that adherence to the Traditional Mediterranean dietary pattern improves asthma and wheeze in children (Papamichael et al, 2017; Castro-Rodriguez et al, 2017) however, more intervention studies are needed to provide concrete evidence establishing the cause-effect relationship.  The possibility that small dietary changes could reduce or prevent asthma and other inflammatory disease symptoms indeed are fascinating. Furthermore, in clinical practice management of asthma patients consists of pulmonary function tests and presence/absence of symptoms and does not include dietary evaluation. On a universal scale, no dietary recommendations are available for asthma patients. Therefore, the findings of this manuscript add valuable evidence to the gap in the literature and set the foundation for dietary guidelines to be provided by health professionals involved in the primary care of pediatric asthma patients.

Although the authors have made a good effort in presenting their findings, there are a number of concerns that need to be clarified. More specifically, there are typographical and grammatical errors that require rectification. Throughout the manuscript, the authors refer to their own studies. More studies conducted by other researchers are warranted to support their findings. In the discussion section, a paragraph on the study strengths/limitations as well as conclusion should be added to link the content.  Please refer to my comments to the authors below. See this as an opportunity in mastering the art of scientific writing.

================================================

Comments to the Authors:

-Methods:

-Comment 1

Line 128:It is worth noting, in particular, the presence of fruits…..’

An important characteristic of the TMD is the ‘abundance’ of seasonal, locally grown fruit and vegetables. I think that the high intake of FV should be highlighted in the text.

Comment 2

-In table 6, fish should be included as a component of the TMD

Comment 3

-Line 144 (hs-CRP, TNF-alpha, IL-1, IL -6, C-3 and C-4)

Add a section labeled ‘abbreviations’ defining these abbreviations.

Comment 4

-Line 148. Why was sample size not estimated using a tool such as G*Power Analysis (Faul, 2007)? How was the sample of 1200 derived?

Faul, F., Erdfelder, E., Lang, A.-G., & Buchner, A. (2007). G*Power 3: A flexible statistical power analysis program for the social, behavioral, and biomedical sciences. Behavior Research Methods39, 175-191.

https://www.psychologie.hhu.de/arbeitsgruppen/allgemeine-psychologie-und-arbeitspsychologie/gpower

Comment 5

- In the statistical analysis paragraph state how data is presented as frequencies, percentages, means (sd) etc  and indicate below tables what data is shown

For example Table 7, what is presented % or counts

Below Table 9, indicate ‘continuous data are shown as means ± S.D and categorical as percentages or indicate this in the statistical paragraph. In tables, state which statistical test was used to derive p-values.

-Table numbering. If the tables in the annex are to appear in an online supplement and not in the main manuscript, then these tables should be relabelled as Table S1, Table S2…….. etc and Table 6 in the main manuscript should be labeled Table 1,  Table 7 changed to Table 2 and Table 8 changed to Table 3 etc

Comment 6

-Line 173- on demand consultations? A more appropriate word would be emergency visits for medical care.

-Line 174 Scheduled or control consultations? Perhaps a better word would be usual consultations

Comment 7

-Figure 3 add a legend on the y-axis to indicate what parameter is being shown.

Comment 8

-Line 193 Indicate in text what scale represents optimum MD score ( 8-12)?

Comment 9

-Figure 4 add % to data. Also typographical error ‘Traditional Mediterranea diet’, you mean Mediterranean

Comment 10

-Table 10 Typo error for p-values.  Replace commas with decimal points. Change the mauve font to black bold and beneath table 10 include the statement 'Statistically significant p-values are indicated in bold'

-Table 11, add p-values to indicate that there were significant differences in laboratory tests between the groups

Comment 11

-Line 254-257. Not comprehensible ‘However, with nutritional intervention, we have obtained beneficial ……since the IRD episodes rapidly progressed towards cure, preventing many of these patients from having prolonged pharmacological treatments or undergoing surgery’

 Progressed towards cure? Not appropriate English. Better to say ‘since IRD episodes decreased, there was less need for pharmacological treatments or surgery.’

Comment 12

-line 258   typo error CRBC?  Do you mean RCBC?

Comment 13

-Line 282 ‘By reducing the consumption of processed foods and those with a high glycaemic index, together with an excess of foods of animal origin, we have also observed a decrease in the incidence of overweight and obesity’

Apart from self-citation include refs to other studies showing improvements in IRD after adherence to med diet or a similar healthy diet.

Comment 14

-Line 292-295. ‘It would have been interesting to have applied TMD to mothers during pregnancy and lactation since antigenic substances from foods that are not typical of TMD could have reached them through the placenta or breast milk’

Support your statement. Add a ref to a study showing the beneficial effect of TMD during pregnancy and lactation on IRDs, or childhood asthma

Comment 15

-Line 295-297 ‘Childhood asthma (RW) is generally triggered by viral infections, which stimulate a previously pro-inflamed or hyper-reactive mucosa.

Add a reference  

Comment 16

Line 298: ‘For this reason, inhaled corticosteroids are used for prolonged periods

to nullify or control this undue inflammatory response’

Nullify inflammatory response? More appropriate to say dampen or downregulate the inflammatory response

Comment 17

-Line 303 ‘Such food can result in an antigenic overload of the immune system, which takes several years to resolve an’’’

Such foods? List specific foods causing antigenic response and add refs

Comment 18

-Line 306-308’ Most of the patients with IRD studied in recent years had positive inflammatory markers, especially TNF-alpha and, to a lesser extent, hsCRP, IL-1, IL-6, C-3 and C-4’

Add a reference.

Comment 19

-Line 317 ‘Industrial juices’ Incorrect English. Better to say ‘commercial fruit juices’.

Comment 20

-Lines 318-321 ‘In general, the glycaemic index was not considered. In the lipid section, the consumption of saturated fats is not limited nor was its consumption evaluated, which does not allow us to detect an excess in the consumption of animal proteins. No assessment was made of the consumption of raw food nor was the minimum amount to be taken specified. Serving sizes and schedules were also not considered. ‘

These statements should be included in a separate paragraph discussing the strengths and limitations of this study. One could commence the paragraph by saying ‘This study should be considered in light of its strengths and limitations’

Comment 21

-Line 330-333 The excess of "antigenic load" caused, above all, by alteration of the microbiota and the increase in proteins of animal origin, could be the cause of the imbalance of the immune system and its excessive enlargement, ultimately causing pathological hyperplasia’

Add a reference

Comment 22

-Line 334-337. ‘Conventional medicine focuses on symptomatic and etiological treatment, forgetting or underestimating the body's ability to achieve balance as soon as the requirements it needs are given. All this translates into an excess of drugs and surgical interventions, which try to eliminate the symptoms and silence and even disavow the work that the body does in its attempt to get rid of what is harming it.’

In the context of the topic of interest, these statements are not relevant. It is not clear what you are referring to. Do you mean that the prophylactic and therapeutic potential of the TMD as a means of symptom control in IRD and in enhancing the functioning of the immune system?

Comment 23

Line 340- assimilable? Do you mean metabolized or assimilated

Comment 24

-lines 349-359 ‘Our research study can be easily replicated in a pediatric consultation, provided that nutritionists are present. It is precisely the lack of nutritionists and the paucity of attention paid to nutrition that explains why the importance of diet in the development of acute diseases and IRD is ignored or obviated. We would have liked to carry out our studies with a control group. However, this was impossible because our child population was being taught to adhere to TMD and it did not seem appropriate to promote the standard diet of Western civilization, since we consider that it is precisely this diet that is the precipitating factor in the development of IRD. We also did not have sufficient funding to perform tests to measure the response of the inflammatory system and the modification of the intestinal flora. It would also have been interesting to monitor expectant mothers and those who breastfeed their children.’

This paragraph should go in the strengths/limitations section.

Comment 25

-Lines 360-370 obviously form the conclusion, but it should link to the previous paragraph.

Lines 360-370 ‘According to our results, many of the foods offered in "Western civilization" have a potentially high capacity to precipitate an undue inflammatory response by raising inflammatory markers and by precipitating the appearance of clinical symptoms compatible with alterations of the immune system. Limitation of suspicious foods and application of the TMD have led to a considerable improvement in clinical symptoms and the normalization of inflammatory markers. We consider excess protein of animal origin and precooked foods as the main suspect foods. Cow's milk and other foods of animal origin could be implicated in the development of childhood asthma. Application of the Traditional Mediterranean Diet from birth, and the limitation of foods foreign to this diet and promoted by "Western civilization", could significantly reduce the incidence of infant morbidity and the disappearance of IRD, including childhood asthma.’

Better to say “ In conclusion, the findings of this study suggest that many foods offered in “Western civilization”……………………childhood asthma”

Comment 26

-Line 370. ‘Disappearance of IRD’, better to say ‘reduction in IRD, including childhood asthma'

Comment 27

-Annex Table 5, typo error in the footnote below the table ‘Adaptinf’.

You mean Adapting?

Author Response

Consulte el archivo adjunto

Reviewer 2 Report

In the article" Effects of the mediterranean dite od morbidity from inflammatory and recurrent diseases with special reference to childhood asthma" the authors evaluated incidence of of recurrent and inflamatory respratory diseases and asthma in children after 8 years of condacting a mediterranean diet.

Unfortunately without comparison with the control group the authors could not draw the above conclusions. Due to a serious methodological error, the interpretation of the results is very dubious.

The authors put a lot of effort and collected data on the effect of the Mediterranean diet on the frequency and treatment of respiratory diseases and asthma over 8 years.

Unfortunately, only in infant group subjects were monitored the differences between the two groups of subjects on different diets. Therefore, it is suggested that the authors "use" these results and write a new publication.

Author Response

Consulte el archivo adjunto

Round 2

Reviewer 2 Report

In a revised article of changed title; Effects of the Mediterranean diet on morbidity from inflammatory and recurrent diseases with special reference to childhood asthma the authors introduced the required modifications and thus improved the article.

Therefore, I am of the opinion that this version of the article can be accepted for publication.

This manuscript is a resubmission of an earlier submission. The following is a list of the peer review reports and author responses from that submission.

Round 1

Reviewer 1 Report

Comments to the Authors

Summary

This manuscript reported the findings of an intervention study undertaken to evaluate the efficacy of the Mediterranean diet on morbidity from inflammatory and recurrent diseases (IRD) including childhood asthma. According to the investigators, adherence to the Mediterranean dietary pattern reduced significantly IRD prevalence and infant morbidity including recurrent infections,  asthma exacerbations as well as antibiotic use and asthma medication, emergency visits for medical care, and need for surgical intervention. Indeed, the therapeutic benefits of the Mediterranean diet in chronic disease, as well as pediatric asthma, are well-known. Given that long-term medication use can cause health complications in children, it is worthwhile for clinicians to explore whether dietary modifications can prevent or at least be used as an adjunct to conventional pharmacotherapy. The current manuscript reviewed would be of interest to health professionals involved in the care of pediatric patients suffering from IRD and asthma.

Overall, the manuscript is well-written and comprehensible for a wide range of readers including non-medical researchers.  However, there are a couple of concerns that require clarification and typographical errors.

Main manuscript

-Line 102 ‘Concerning morbidity, we have defined IRD as the repetition of the same inflammatory pathology in more than three episodes in 6 months or four episodes in 1 year’

A reference is needed to justify the criteria used for IRD

-Line 172 Sample size

Describe how sample size was calculated based on which parameter, the test used, and software.

-Line 208 Table 9, indicate below the table that data are shown as means and standard deviations as well as the test used to derive p-values.

-Line 230, figure 4 pie chart for 1-year follow-up.

Typographical errors require rectification low cuality diet –you mean Low-quality diet;

Neet to improbe- need to improve

-Line 242, Table 10, font size not consistent. Indicate below the table, what group 1 and group 2 refer to and that data is presented as means, standard deviations, and which statistical test was used to derive p-values. Please be consistent with p-values, check journal guidelines. Otherwise, according to APA standards, p-values ≥ 0.05 should be shown to 2 decimal places and < 0.05 to 3-decimal places.

-Annexe 1. Have the questionnaires (TMD test, TMD-B test) designed for the purpose of this study been validated previously?

-Line 601, typo error *Always adpatinf – do you mean Always adapting?

Author Response

Por favor vea el adjunto

Reviewer 2 Report

Comments:

The article “Effects of the Mediterranean diet on morbidity from inflammatory and recurrent diseases with special reference to childhood asthma” indicate that traditional mediterranean diet ( TMD) reduce the incidence of infant morbidity and contributes to the disappearance of inflammatory and recurrent diseases (IRD). Moreover, western foods with high antigenic power could be involved in the appearance of IRD. The study is presented in systematic way and interesting for reader group. Few suggestions that could help to improve the quality of the manuscript:

Comment 1: Legumes are integral part of mediterranean diet and several legumes (beans, lentils, and pea) are reported to show allergic manifestations. Many IgE binding proteins are found in legumes that provoke allergic diseases. What is authors’ opinion?

Comment 2: Line 168-169: Authors measured several inflammatory markers for examination of IRD. Eosinophils are playing critical role in disease progression especially allergic manifestations. Did authors examine the IL-4, IL-5, IL-13, IL-18 cytokines in the patients group and in recovery group?

Comment 3: Line 311-313: “By reducing the consumption of processed foods and those with a high glycaemic index, together with an excess of foods of animal origin, we have also observed a decrease in the incidence of overweight and obesity”.  In western countries, obesity is leading cause of liver related complications and inflammation. Did authors’ measured the serum AST and ALT level before and after introduction of TMD in patients group?

Comment 4:

Few minor errors:

Line 181: ‘Initially, all those who had had some…’ remove extra “had” from sentence.

Figure 4: Change the graph color for better differentiation of groups.

1 year graph:  “low cuality diet” replace with “low quality diet”

“neet to improbe” replace with “Need to improve”

.

Author Response

Por favor vea el adjunto
